# Human Remains Identification Using Micro-CT, Chemometric and AI Methods in Forensic Experimental Reconstruction of Dental Patterns after Concentrated Sulphuric Acid Significant Impact

**DOI:** 10.3390/molecules27134035

**Published:** 2022-06-23

**Authors:** Andrej Thurzo, Viera Jančovičová, Miroslav Hain, Milan Thurzo, Bohuslav Novák, Helena Kosnáčová, Viera Lehotská, Ivan Varga, Peter Kováč, Norbert Moravanský

**Affiliations:** 1Department of Stomatology and Maxillofacial Surgery, Faculty of Medicine, Comenius University in Bratislava, 81250 Bratislava, Slovakia; bohuslav.novak@fmed.uniba.sk; 2Institute of Forensic Medical Expertise, Expert institute, Boženy Němcovej 8, 81104 Bratislava, Slovakia; pkovac@gmail.com; 3Department of Simulation and Virtual Medical Education, Faculty of Medicine, Comenius University in Bratislava, Sasinkova 4, 81272 Bratislava, Slovakia; helena.svobodova@fmed.uniba.sk; 4Department of Graphic Arts Technology and Applied Photochemistry, Institute of Natural and Synthetic Polymers, Faculty of Chemical and Food Technology, Slovak University of Technology in Bratislava, Radlinského 9, 81237 Bratislava, Slovakia; viera.jancovicova@stuba.sk; 5Institute of Measurement Science, Slovak Academy of Sciences, Dúbravská Cesta 9, 84104 Bratislava, Slovakia; umerhain@savba.sk; 6Department of Anthropology, Faculty of Natural Sciences, Comenius University in Bratislava, 84215 Bratislava, Slovakia; milanthurzo@gmail.com; 7Department of Genetics, Cancer Research Institute, Biomedical Research Center, Slovak Academy of Sciences, Dubravska Cesta 9, 84505 Bratislava, Slovakia; 82nd Department of Radiology, Faculty of Medicine, Comenius University in Bratislava, Heydukova 10, 81250 Bratislava, Slovakia; viera.lehotska@ousa.sk; 9Institute of Histology and Embryology, Faculty of Medicine, Comenius University in Bratislava, 81372 Bratislava, Slovakia; ivan.varga@fmed.uniba.sk; 10Institute of Forensic Medicine, Faculty of Medicine Comenius University in Bratislava, Sasinkova 4, 81108 Bratislava, Slovakia

**Keywords:** dental pattern, forensic dentistry, forensic radiology, forensic chemistry, machine learning, identification, dissolution, sulfuric acid, dentition, teeth, acid degradation

## Abstract

(1) Teeth, in humans, represent the most resilient tissues. However, exposure to concentrated acids might lead to their dissolving, thus making human identification difficult. Teeth often contain dental restorations from materials that are even more resilient to acid impact. This paper aims to introduce a novel method for the 3D reconstruction of dental patterns as a crucial step for the digital identification of dental records. (2) With a combination of modern methods, including micro-computed tomography, cone-beam computer tomography, and attenuated total reflection, in conjunction with Fourier transform infrared spectroscopy and artificial intelligence convolutional neural network algorithms, this paper presents a method for 3D-dental-pattern reconstruction, and human remains identification. Our research studies the morphology of teeth, bone, and dental materials (amalgam, composite, glass-ionomer cement) under different periods of exposure to 75% sulfuric acid. (3) Our results reveal a significant volume loss in bone, enamel, dentine, as well as glass-ionomer cement. The results also reveal a significant resistance by the composite and amalgam dental materials to the impact of sulfuric acid, thus serving as strong parts in the dental-pattern mosaic. This paper also probably introduces the first successful artificial intelligence application in automated-forensic-CBCT segmentation. (4) Interdisciplinary cooperation, utilizing the mentioned technologies, can solve the problem of human remains identification with a 3D reconstruction of dental patterns and their 2D projections over existing ante-mortem records.

## 1. Introduction

Human remains are relatively easy to destroy. However, some tissues, such as bones and teeth, are more durable than others. Teeth are still not as durable as some dental materials, which can withstand not only mechanical forces but also the heat of a fire, decomposition, or other means of destruction, including the impact of acids. Acid destruction is often the preferred method to prevent body identification. It can dissolve DNA, metals, and the whole body, and liquefy even bones and teeth. Sulfuric, hydrochloric, and nitric acids, and their combinations, are among the most frequently used acids for this purpose [1,2,3,4,5,6]. In fact, after extensive destruction, as in with acid, the reconstruction and identification of human remains are extremely difficult, and even the most modern technologies still do not offer a textbook method for a virtual re-association of fragmented human remains [2,7,8,9,10,11,12].

Identifying an unidentified body is seldom a straightforward path, and forensic dentistry has long been a reliable discipline in the process of human identification [1,13,14]. Dental characteristics, such as the morphology of a tooth, differences in size and shape, number of cusps, color, dental restorations, pathologies, hypodontia, wear patterns, malocclusions, and other distinctive dental anomalies, give every individual a unique identity—a dental pattern. The utilization of dental patterns for human remains’ computerized identification is a common and undisputed method in forensic dentistry [2,8,9,11,15,16]. Forensic experts worldwide continue to remain divided about the need for a definition of a minimum number of concordant points to confirm dental identification [17]. There seems to be no basis for defining this minimum, and numerous studies reinforce the view that each case has its own individuality and, thus, should be treated as such [17,18,19,20,21,22,23]. Teeth are the most durable structures of a body. They can resist various means of destruction more than any other skeletal tissue, albeit commercially available acids, which can be used to obliterate them in an effort to prevent their identification. Criminals may use sulfuric acid, HNO_3_, or HCl to hinder the person’s identification; however, individuals can be identified using resilient dental patterns. These frequently persist after exposure to concentrated acids [24,25]. Teeth can serve as a reliable identification tool even after eight hours of exposure to concentrated sulfuric acid (H_2_SO_4_); however, they completely dissolve in concentrated HCl and concentrated HNO_3_ after eight hours. Very few materials of a specific composition—most often plastic composites—usually avoid total destruction. These can be dental materials, such as root or composite fillings, plastic nails, or other composite materials. Dental materials frequently resist decomposition and high temperatures and are among the last ones to disintegrate after death. The principles of dental identification lie in the fact that no two dental restorations are the same, and teeth and their therapeutic modifications are unique to the individual. The concept of the dental patterns described above takes advantage of the fact that the therapeutic alterations of teeth and their surrounding tissues (tooth fillings, crowns, bridges, implants, and others) can be even more resilient to destruction than the original dental tissues [2].

Post-mortem-computed tomography is now routinely applied, and a CT scanner is a valuable tool for post-mortem forensic examinations [26]. It is an established documentation tool in forensic medicine due to its diagnostic accuracy [27]. When the problem is described as morphological tissue destruction, and experts face the necessity of matching tissue to dental records, CBCT is the most typical computed tomography available. It is used primarily in maxillofacial applications and may also be particularly useful in some forensic contexts, offering numerous advantages for post-mortem forensic imaging, including its good resolution for skeletal forensic radiology, its relatively low costs, and simplicity [28]. The identification of unknown remains, by means of a dental X-ray comparison, is well-established in dental forensics. Two-dimensional (2D) intraoral photos, or 2D X-ray images from dental records, are most frequently used for matching. If many individuals need to be identified in a brief time period, dental identification can be a time- and resource-consuming procedure [29]. Protocols have been developed to reproducibly and reliably reformat the CBCT volumes. The reformatted images can be compared directly with conventional digital images from the same anatomic area. Images that are derived from the CBCT volumes are similar enough to those from conventional dental radiographs and allow 2D dental forensic comparisons and identification. CBCT offers a superior option over multi-slice CT scanners for this purpose [30]. Given that a 2D reconstruction is appropriate for matching two 2D radiographic images, it is difficult to find the appropriate projection. As dental records are typically in 2D format, it is sufficient, albeit not necessary, to give up on the advantage of 3D-reconstructed matching to the 2D image. The reconstruction of skeletal remains using conventional CBCT provides accurate 3D reconstructions, which demonstrates its reliable use as a forensic tool [31]. The micro-CT has currently limited availability within forensics practice. This paper places micro-CT in context, within the presented complex analysis, in an effort to reconstruct dental patterns that face problems with morphological destruction of tissues [32].

The application of chemistry and its subfield, forensic toxicology, in a legal setting, defines forensic chemistry. The identification of unknown materials found at a crime scene can be processed by a forensic chemist. In the focus of this paper, the bone and tooth tissues, including dental fillings, undergo particular chemical changes when exposed to 75% sulfuric acid. Specialists in this field use a wide array of methods and instruments to help identify the chemical composition of unknown substances. One of these technologies, implemented in this research, is attenuated total reflection (ATR) in conjunction with Fourier transform infrared spectroscopy (FTIR), and is one of the key pillars of the introduced novel hybrid and complex forensic approach for the reconstruction of dental patterns. Numerous studies on the changes to bone chemistry have used FTIR spectroscopy in conjunction with other chemometrics methods [33], according to Chophi et al., 2020 [34].

Dental materials exposed to chemically or otherwise destructive forces undergo dynamic processes that might stop on a particular level, and even the aging of dental material might render chemical changes. For example, in methacrylate-peak determination, ATR-FTIR can be utilized to investigate the polymerization of dental-methacrylate mixtures. Numerous dental studies investigate polymerization kinetics with ATR-FTIR utilization. However, the peak selection techniques used to determine monomer–polymer transformation can differ, subsequently altering the results. ATR-FTIR can reliably evaluate the polymerization of methacrylate and is, therefore, suitable for forensic analysis, as well [35].

The advanced AI algorithms have proven to be useful tools in processing digital data and will soon subdue the areas of automatized assessment for age, sex, ethnicity, microfractures, or dental pattern identification and comparison. This is inevitable and will provide forensic expertise with unprecedented and efficient analytic tools [36]. Post-mortem dental patterns that are reconstructed, possibly with AI algorithms, can be compared to considerable amounts of digital records of variable quality, covering the anatomical region, time of origin, or type of diagnostic technology. Currently, the technologies providing data that can contribute to the final completion of the mosaic include the 3D intraoral optical scan, 3D-CBCT skull scan, 3D nuclear magnetic resonance head scan, 3D lidar face scan, 3D ultrasonographic scan of the submandibular region, 2D dental local X-ray scans or OPG scans, and many more. Most of them are dismissed today because they are too difficult to implement, and research has not yet proven them to be accurate for the final matching. This wide variety of possibly hundreds of medical and dental records is also randomly distributed, in the time from childhood to the last days. The ability of AI to simulate aging, based on large data-trained networks, can help to adapt these records in a set time and valuably incorporate them into the final mosaic of the reconstructed dental pattern [36]. In contrast to the conventional approaches, AI methods are useful in gaining extraordinary classification correctness without the necessity for an extremely precise tooth segmentation.

This article aims to introduce a novel multidisciplinary approach for the forensic human identification of acid-obliterated remains based on dental pattern reconstruction. This paper investigates the benefits of an interdisciplinary combination of digital photography [7,37,38], cone-beam computed tomography (CBCT) [39,40,41,42], micro-computed tomography (micro-CT) [43], advanced AI algorithms, such as the 3D convolutional neural network (CNN) algorithms [36,44,45,46] and attenuated total reflection (ATR) in conjunction with Fourier transform infrared spectroscopy (FTIR) [47,48].

The highlights of this article are:A human mandible with teeth (treated post-mortem) was degraded in 75% sulfuric acid, and the accompanying morphological and chemical changes were documented;CBCT and micro-CT technologies were used for the 3D reconstruction of dental patterns and descriptive morphological evaluation;ATR-FTIR spectroscopy was utilized to investigate the changes in dental restorations;The advanced AI–CNN algorithm was utilized for automated mandible segmentation;This research provides an unprecedented 3D morphological set of four stages of degradation of human mandibular bone and teeth presented in five different regions.

## 2. Materials and Methods

The primary working hypothesis for this research was that dental restorations could resist the degradation effect of 75% sulfuric acid more than dental tissues.

The secondary hypothesis was premised that 3D-reconstructed dental patterns can be utilized for matching with 2D dental records represented by an intraoral digital photo and OPG scan.

### 2.1. Materials

#### 2.1.1. The Bone of Human Mandible with Teeth

The bone of the human mandible with teeth (Figure 1a,b), described in this article, was donated from the Institute of Forensic Medical Expertise—Forensic.sk. It originated as a random anonymous finding during an excavation of a cancelled cemetery and was intended for disposal. It was found without a remaining skeleton and collected by forensic experts for scientific research more than 20 years ago.

#### 2.1.2. Various Dental Materials

Dental fillings of a special shape were created in three pairs of teeth with three different dental materials (Figure 2a). Older dental fillings were removed. The objective was to create a plane surface that would extend the surrounding enamel surface to allow contact with the FTIR sensor (Figure 2b). Later, the objective of the FTIR examination of dental fillings was abandoned due to possible breakage of the whole tooth in the later stages of preparation for the degradation process. Three pairs of tooth fillings were created from the following dental materials:Dental amalgam—Ana 2000 capsules non-gamma-two, extra-high copper (containing 43% silver, 26.1% copper, 30.8% tin);Glass ionomer—GC FUJI IX GP wear-resistant multipurpose (containing powder: 95% Fluro alumino silicate glass, 5% Polyacrylic acid powder; liquid: 40% Polyacrylic acid and tartaric acid, 50% distilled water, 10% Polybasic carboxylic acid);Dental composite—Neo Spectra ST (containing methacrylate-modified polysiloxane, dimethacrylate resins, fluorescent pigment, UV stabilizer, Camphorquinone, Ethyl-4 (dimethylamino)benzoate, Bis (4-methyl-phenyl) iodonium hexafluorophosphate, Barium–aluminium–borosilicate glass, Ytterbium fluoride, iron-oxide pigments and titanium-oxide pigments, according to shade). Prime and Bond Universal were used as an adhesive system in the dental filling/restorations with a composite.

### 2.2. Methods

The principles for the digital matching of the 2D representation from the panoramic X-ray to the 3D-CBCT representation of a particular dental filling followed these rules:The 2D shape of each dental filling was extracted graphically from the panoramic X-ray 2D image;The outer contours and inner contours with shades of gray gradient were identified, thus creating a unique grayscale image pattern with contours;The 3D-CBCT dental fillings were segmented and extracted from the tooth;The semitranslucent 3D model from the CBCT of each dental filling was transposed over each 2D unique grayscale image pattern with contours in supposed positions;In the basic match, a viewer’s perspective was adapted to achieve an exact match.

### 2.3. Digital Optical Scanning

Digital optical scanning of the original mandible was performed with the iTero Element intraoral scanner (Align Technology, Inc., San Jose, CA, USA) and exported to STL (Figure 3a,b). Models from this output were compared to the segmented models from the CBCT to prove this optional step as a valid procedure. Only one initial scan was created and was sufficiently accurate in comparison to the segmented CBCT mandibular surface (Figure 3b).

### 2.4. CBCT Scanning

CBCT was used for scanning (Figure 4a,b) and was performed with the Planmeca ProMax 3D Mid CBCT (Planmeca Group, Helsinki, Finland), with the following parameters:-A panoramic exposure with the following settings and values: 2D panoramic, standard, patient size M = medium-sized adult, 67 kV, 11 mA, 15 s;-The 3D exposure was performed with the following settings: CBCT volume Ø100 × 100 in a high-definition (HD) mode, voxel size 150 μm;-Two pairs of OPG and CBCT were created. The first scan was before the teeth preparative treatment, and the final scan was after the dental filling/restoration.

The final CBCT scan, with the custom post-mortem dental filling/restoration, was subjected to a manual as well as a novel advanced AI–CNN algorithm for the segmentation of the teeth and jaw.

### 2.5. STL Segmentation

For the manual segmentation, we used a full version of the software, Invivo dental 6.5 (Anatomage, Inc., Santa Clara, CA, USA). For the AI-automated segmentation, the mandibular segment, teeth, dental sockets, and canal for the mandibular nerve, were segmented with the Diagnocat online program (LLC “DIAGNOCAT” Moscow, Russia) (Figure 5). This solution then processed the CBCT scan of the mandible within five minutes and created the STL segments of the described anatomical structures with the use of a CNN [44,49,50,51]. The STL segments were downloaded from this program and stored on a computer.

### 2.6. Micro-CT Scanning

After the CBCT scanning, the mandible was cut into five samples (Figure 6) and scanned initially with X-ray and micro-CT.

The micro-CT scans were created using a microtomography system, Nanotom180 (GE Healthcare Inc., Phoenix, AZ, USA). The X-ray tube was equipped with a tungsten target, and the micro-focusing mode, M0, was applied. For the standard micro-CT scan of a tooth, the following settings were used: accelerating voltage U = 150 kV, beam current I = 80 µA, and detector sampling time t = 750 ms. The voxel size was twenty µm, and a total of 1800 X-ray projections were recorded during one CT scan. For the filtering of the outgoing X-ray beam, a 0.2 mm Cu plate was used.

For the micro-CT scans of a tooth with an amalgam filling, different settings were used: accelerating voltage U = 170 kV, beam current I = 90 µA, detector sampling time t = 500 ms, and pixel binning 2 × 2 was applied. For the filtering of the outgoing X-ray beam, a 0.5 mm Sn plate was used. The 3D-volume reconstruction was done using the Phoenix datos|x CT software using the Feldkamp filtered back-projection algorithm. The 3D datasets were rendered and segmented using the VGStudio MAX 2.1 software (Volume Graphics GmbH, Heidelberg, Germany). For the segmentation of volume data, the region-growing method was mostly used, along with the opening/closing and erosion/dilation image processing techniques (Figure 7a–c). The segmented 3D model of the preserved dental filling is a crucial part of the dental pattern and was used for matching it to different 2D projections from the pre-mortem dental records (Figure 7c).

### 2.7. Acid Exposure

The five segments of the mandibular bone with treated and untreated teeth were exposed to 75% sulfuric acid in sequential exposition after 2, 6 and 24 h. After each exposition, the specimens were washed and dried, and a micro-CT scan was performed on every sample. There were four micro-CT scans for each sample, and each scan was segmented into a bone part and tooth part as a separate STL (Figure 8). A set of output STLs were made publicly available online. The visualizations of the micro-CT-segmentation outputs of all five samples are summarized in detail in the Appendix A section of this article, as well as the digital photo documentation.

After the timed exposure, all five samples were taken from the acid bath and washed three times in deionized water and then dried in a multifunctional oven, APT Line Series FED, with the regulation R 3.1 by Fisher Scientific (Waltham, MA, USA), at a temperature of 40 °C for 60 min. At this time of intermission, a micro-CT scan was performed after 2, 6, 24 and 96 h. Additionally, the ATR-FTIR spectra were measured; however, only after 2, 6 and 24 h. The ATR-FTIR was not measurable after 96 h of acid exposure as the samples were too fragile, and the ATR-FTIR measurement would represent a significant risk to the sample’s mechanical destruction.

Subsequently, after the measurements, the samples were re-immersed in fresh/new sulfuric acid (75% H_2_SO_4_), and the whole procedure was repeated.

### 2.8. FTIR Spectroscopy

The ATR-FTIR spectroscopy method was performed with the following settings: The attenuated total reflectance (ATR) mode for the FTIR spectra of bone and teeth samples was acquired using the FTIR spectrophotometer, Varian Excalibur Digilab FTS 3000MX (Palo Alto, CA, USA), equipped with the ATR adapter with the diamond crystal. The measurement range was from 4000 to 600 cm^−1^; resolution was 4 cm^−1^; sensitivity was 8; with thirty scans. The background was air. Three different spectra, in various places, were measured, averaged, and evaluated using the Origin (version 8.5, MicroCal) software (Malvern Panalytical Ltd., Malvern, UK).

## 3. Results

### 3.1. Descriptive Morphological Evaluation Based on Micro-CT Analysis

A significant volume loss was observed on the segments of bone, teeth, and glass-ionomer cement. An insignificant volume loss was detected on amalgam (<0.1%). No volume loss was detected on the composite dental material. The volumetric changes after 6, 24 and 96 h are visualized in Figure 9 and detailed in Table 1.

Our results showed an outstanding resistance by the high-copper non-gamma-two amalgam and composite resin to the impact of 75% sulfuric acid. Even after 96 h in the 75% sulfuric acid bath, the volumetric loss was less than approximately 0.1% in amalgam and none in the composite (Table 1 and Figure 9).

### 3.2. Digital Matching and AI Implementation in CBCT Segmentation

The digital matching of the 3D-reconstructed dental fillings, segmented from micro-CT, which survived the acid bath, was successfully performed on all four localizations (2× amalgam, 2× composite). The restorations with glass-ionomer cement dissolved and were not reproducible. One exemplary match via the digital OPG was visualized (Figure 10). With the digital matching, after the basic 3D to 2D matching, the viewer’s perspective was adapted to achieve an exact match. This was successful with both the amalgam and composite restorations (4).For example, for the 3D model from the CBCT (Figure 10 red) and a unique grayscale image pattern with contours (Figure 10 gray). Semitranslucent 3D models of four experimental fillings (2× amalgam, 2× composite) were overlaid in all possible sixteen combinations with three independent operators. In all four identified matches, no discrepancy was found between the observed pair-alignment. Even after the primary basic match, no doubts arose about the 2D–3D pair allocation. Despite this, after the primary match, a fine-tuning of the 3D-model perspective adjustment was performed. We achieved a perfect match in all four cases, with the level of evaluators’ confidence approaching certainty (Figure 10 semitranslucent 3D model—lower row).

The optional objective of comparing the manual (software) and AI-segmentation methods of CBCT was fulfilled. This was possibly the first laboratory implementation of advanced AI algorithm utilization in forensic research. The result of the successful automatic AI segmentation lasted less than 5 min and provided comparable results with the manual segmentation, which took the human operator more than 2 h. Small errors were observed in the nerve canal segmentation and the distal part of the terminal molar contours (Figure 5b).

### 3.3. ATR-FTIR Spectroscopy

ATR-FTIR spectroscopy was used to evaluate the chemical changes of the bone due to its degradation by 75% sulfuric acid. The spectrum of the original sample (Figure 11a) contains characteristic bands of both organic and inorganic bone components. Amide I band at 1635 cm^−1^, amide II band at 1540 cm^−1^, as well as the weak bands at 3080 cm^−1^ (Amide B) and 1240 cm^−1^, are characteristic of collagen, which is a major organic component of bones [52]. The absorption band at 1020 cm^−1^ belongs to the phosphate group of hydroxyapatites (bioapatite, Ca_5_(PO_4_)_3_OH), the main inorganic component of bone, and the bands at 1460, 1418 and 874 cm^−1^ belong to the carbonate group in CaCO_3_ [53].

After immersion of the bone in 75% sulfuric acid, the bands dropped after two hours, mainly in the amide region, which indicated a preferential removal of collagen from the bone surface. Significant bands were formed at 1100 and 1150 cm^−1^ (typical of the sulphate group in the calcium sulphate), which indicated the formation of a layer of calcium sulphate on the surface of the samples. Since the beam penetration depth of our ATR-FTIR setting was 1–4 mm (1 mm at 4000 cm^−1^ and 4 mm at 600 cm^−1^), we measured only a thin surface layer, and the dominant bands belonged to the calcium sulphate forming on the surface after a brief time. This complicated the process of evaluating the changes in the material and limited us to measuring surfaces. The measurement of ATR-FTIR after more than 24 h failed to perform, due to the mechanical disruption of degraded bones due to pressure. The accumulation of calcium compounds on the bone’s surface was also confirmed by the X-ray fluorescence (XRF) measurements, where, after a brief time, there was an increase in the calcium content compared to phosphorus. XRF is a non-destructive analytical method used to define the elemental composition of materials. XRF analysers decide the chemistry of a sample by calculating the fluorescent (or secondary) X-ray emitted from a sample when it is excited by a primary X-ray source.

With the same measurements on the model tooth (Figure 11b), the teeth showed to be more stable and were naturally expected to last longer than bone [48]. The main absorption bands of the tooth were in the same areas as the bone. The band at 1020 cm^−1^ was retained after 6 h and partly after 24 h in sulfuric acid, while the absorption bands belonging to the collagen (1641 and 1540 cm^−1^) decreased significantly, thus leading to a faster loss of organic components. In the spectra measured at different times of acid action, there was not only a simple decrease in the bands, but the differences were larger (e.g., between two and six hours of exposure to 75% sulfuric acid), which is related to the fact that the tooth is a multilayer material and after dissolving the enamel, we could observe dentin (the spectrum after six hours of sulfuric acid treatment corresponds to the spectrum of dentin, where the bands between 1410 and 1650 cm^−1^ are typical for carbonate overlapped with collagen) [54]. As in the case of bone, CaSO_4_ formation was also observed on the tooth surfaces in all the samples.

## 4. Discussion

Dental identification is seldom a straightforward path, and various techniques help to identify human remains in incidents, such as traffic accidents, terrorist attacks, fires, mass murders, natural disasters, and many others [2,12]. The solution to the identification problem lies in a multidisciplinary combination of the most modern digital methodologies that link the forensic fields of anthropology, dentistry, radiology, chemistry and possibly with the implementation of artificial intelligence. The remains can be identified with the help of 3D scanning and 3D reconstruction. A three-dimensional (3D) digital reconstruction of the most resistant parts of human remains—teeth, with their extremely specific modifications based on medical records—provides the possibility of reconstruction of the so-called dental patterns, which define a species-specific arrangement of teeth by type in each quadrant of the mouth. Dental characteristics, such as the morphology of the tooth, differences in size and shape, number of cusps, color, restorations, pathologies, hypodontia, wear patterns, malocclusions and the position of the tooth, and other distinctive dental anomalies, give every individual a unique identity—a dental pattern [2,15].

The results supported our aims and showed the possibility of introducing a novel multidisciplinary approach for the forensic identification of human remains exposed to acid based on the reconstruction of the dental model. The CBCT and micro-CT technologies were used for the 3D reconstruction of dental samples, which were used for the morphological evaluation of the sample, for automated segmentation of the mandible using the AI algorithm, and ATR-FTIR spectroscopy was used to evaluate the chemical changes. An important output of this experiment was the comparison of the AI algorithm’s implementation and automatic segmentation with manual segmentation. The results of the work were comparable, but the automation brought considerable time savings.

The documentation of the morphological and chemical changes showed a loss in the volume of dental materials and bones after exposure to 75% sulfuric acid, with a significant loss observed in the segments of bone, teeth, and glass-ionomer cement. In contrast, little or no volume loss was found in the amalgam and composite dental materials. The findings for the resistance of the high-copper non-gamma-two amalgam and composite resin to the 75% sulfuric acid correlate with the complex research presented in the master thesis of Trapp 2018 from the School of Medicine, Boston University [55,56], which provides comprehensive research on the effects of household corrosive substances on tooth restorations. Particularly, the author provides an evaluation of the effects of the solution containing 93.2% sulfuric acid on silver amalgam (Figure 4.3, page 62 [55]). The silver-amalgam tooth filling had minimal changes and was also well-resistant to sulfuric acid even after 264 h exposure. To approximate these findings, it is important to understand the difference between high-silver and high-copper amalgam. In general, the high-copper amalgam is stronger, and it might be expected to be even more resilient to the sulfuric acid solutions than the high-silver amalgam. However, this resistance to sulfuric acid was not shared with the third-used dental material, glass-ionomer cement. The glass-ionomer dental filling dissolved completely after 96 h. The results suggest that dental restorations from typical ones, such as the high-copper non-gamma-two amalgam and the composite methacrylate resins, have a minimal volume loss and deformation when exposed to 75% sulfuric acid. Thus, they are suitable for the reconstruction of a dental pattern during the identification process. [55,56] The acid concentrations are especially important for the final morphological and chemical impact. This was confirmed by the research by Raj et al. in 2013 [57]. The research evaluated the resistance of teeth to acids (37% conc. HCl, 65% conc. HNO_3_, and 96% conc. H_2_SO_4_). The research confirms that teeth will completely dissolve in HCl and HNO_3_; however, in 8 h, they still retain morphology in H_2_SO_4_. The concentration of the acid is one important aspect, and the time of exposure is another. Other studies [58], conducted earlier by Dr. Mazza and published in the *Journal of Forensic Sciences* [58], were initiated by judges in so-called “mafia” crimes to examine whether the body could be destroyed by immersion in acids and whether any remains could be identified at all. This study aimed to observe the behavior of teeth exposed to four types of acidic solutions. The teeth were placed in plastic jars with 25 mL of acid and observed. Experience has shown that teeth completely dissolve after fourteen hours of immersion in a 37% hydrochloric acid solution; however, even after ninety hours in 96% sulfuric acid, the destruction of the samples was still incomplete. In nitric acid, the teeth completely dissolved in 12 h, and in 17 h in a solution of hydrochloric acid/nitric acid of a 1:3 ratio, it was possible to recognize the characteristic morphological features of dental tissues and structures up to the advanced stages of degradation [58].

Regarding the significant differences in the acid’s impact, depending on its concentration, a group of forensic scientists [59] decided to engage in research to dispel some myths and test whether the body, after soaking it in acid, does or does not disintegrate within a few minutes. Massimo Grillo of the University of Palermo and his collaborators published their findings [60] in the *New Scientist* in 2014. When these researchers placed pieces of pig carcasses into sulfuric acid, it took several days for the meat to dissolve. When water was added to the mixture, they were able to reduce the dissolution time to 12 h for muscle and cartilage and two days for bone. This is a valuable observation. It is known that acid should be poured into water and not the other way around, and in the case of sulfuric acid, a strongly exothermic reaction produces a lot of heat and the fumes of explosive gases. Dutch research [59] confirmed the experience of forensic experts that, in addition to the macroscopic findings, such as those of bone residues and artificial teeth, in both cases, clear microscopic bone residues were found: (partially) digested bone, thin-walled structures and recrystallized calcium phosphate. Although some believe that the body can be completely dissolved in acid, at least some of these microscopic residues are always found. These microscopic residues might be a future path for implementing AI algorithms in the modern reconstruction of dental patterns.

Currently, the quantity of studies describing the utilization of AI in forensic aspects is rather low. Nevertheless, AI developments in prosthodontics are demonstrating feasible applications for automated diagnostics and as an effective classification or identification tool. In the future, AI technologies will likely be used for collecting, processing, and organizing patient-related datasets to provide patient-centered and individualized dental treatment [61]. A wide range of “enablers” for AI clinical implementation in dental diagnostics are already known. The opinion leaders in the field will consider these aspects to foster a further implementation of AI in dentistry [62].

A novel AI system, based on deep learning methods, which was used for the mandible auto-segmentation presented in this paper, was scientifically evaluated in the past (Ezhov et al., 2021) [44]. This research assessed real-time AI performance on the CBCT identification of anatomical landmarks, pathologies, and the general clinical effectiveness when used by dentists in a clinical setting. The study of Ezhov et al. showed that the proposed AI system significantly improved the diagnostic capabilities of participating dentists [44].

The success of the AI system in dental implant planning using 3D-CBCT images has recently been published. The further development of AI systems and their utilization in implant planning will facilitate the work of dentists with a support system for the future of implantology practice [51].

AI has proven efficient, as well, in detecting impacted third molars on CBCT scans. The diagnostic performance of the AI application for evaluating the impacted third molar teeth in CBCT images was clinically evaluated. The AI application indicated high accuracy values in the recognition of impacted third molar teeth and their relationship to surrounding anatomical structures [50]. The effectiveness of AI in the identification of periapical pathosis on CBCT scans is impressive, as well. The diagnostic performance of an AI system that is based on the deep-CNN method to detect periapical pathosis on CBCT images has been evaluated in the past. Research by Orhan et al., 2020, on the deep-CNN system showed the system was successful in detecting teeth and numbering specific teeth. Only a single tooth was not correctly identified [50]. The AI system was able to detect 142 of the total 153 periapical lesions. The reliability of correctly detecting a periapical lesion was 92.8%. The deep-CNN volumetric measurements of the lesions were comparable to those with manual segmentation. The AI systems that are based on deep learning methods can be useful for detecting periapical pathosis on CBCT images for clinical application [49].

This paper has also probably introduced the first practical forensic implementation of an AI algorithm used for the auto-segmentation of the mandible, mandibular teeth, and their sockets, as well as the canal of the mandibular nerve. The AI algorithms will bring unprecedented analytical tools; however, they need to be trained first. For this, a vast amount of data is necessary. In the forensic radiology discipline, a specific algorithm can be created to further help with the identification of distinct microscopic characteristics, thus forming an array of specific topographies in 3D. Such a 3D array would represent a new form of AI—an identified dental pattern—hardly comprehensible to a human brain. AI is advancing in the field of dental radiology and will bring a change in thinking within the next decade [36,45,63,64,65,66,67,68,69,70,71,72]. AI technology is still currently underestimated in its clinical implementations in forensic anthropology, dentistry, radiology, and chemistry. It will undoubtedly shift the paradigm in this area [36,63,64,71,73,74,75,76,77,78]. Digital forensic methods are, today, a modus operandi in contrast to the past when they were considered expensive and required pricey and special equipment.

As a limitation of this study, it might be considered that the research has been performed on an old dried-out skeleton. This is not representative of the considerations for reproducing processes accompanying complete human tissues exposed to acids. Another limitation is, on the contrary, the fact that the dental restorations were only a week old before their exposure to an acid bath and were never exposed to the native environment of the oral cavity. Both aspects might not be relevant for the presented results; however, they should be considered before approximating these results to different environments and situations.

## 5. Conclusions

The principal conclusion of this article is that the combined multidisciplinary effort of experts from forensic dentistry, forensic anthropology, forensic radiology, and forensic chemistry can reconstruct the dental pattern from acid-obliterated human remains and use it for successful identification.

The working hypothesis was confirmed and proved that typical dental restorations from high-copper non-gamma-two amalgam and the composite nanohybrid methacrylate-modified polysiloxane (organically modified ceramic) dimethacrylate resins have a minimal volume loss and deformation under 75% sulfuric acid exposure.

The secondary hypothesis was also confirmed, premising that 3D-reconstructed dental patterns can be utilized for matching with the 2D dental records represented by intraoral digital photos and OPG scans.

An advanced AI algorithm was successfully implemented to perform automatic segmentation of the CBCT scan with separate STL outputs of the mandibular bone, teeth, and nerve canal.

This paper described a combination of three technologies with an additional presentation of the AI algorithm for the automated CBCT segmentation and digital photo comparison. This new approach to the reconstruction and comparison of dental patterns depends on the CBCT, micro-CT and chemometric methods. These may be extended in the future by chemometric and micro-CT evaluation of other dental materials. CBCT, micro-CT and ATR-FTIR spectroscopy represent a suitable combination of technologies for a successful morphological and chemometric reconstruction of dental patterns [30,79].

## Figures and Tables

**Figure 1 molecules-27-04035-f001:**
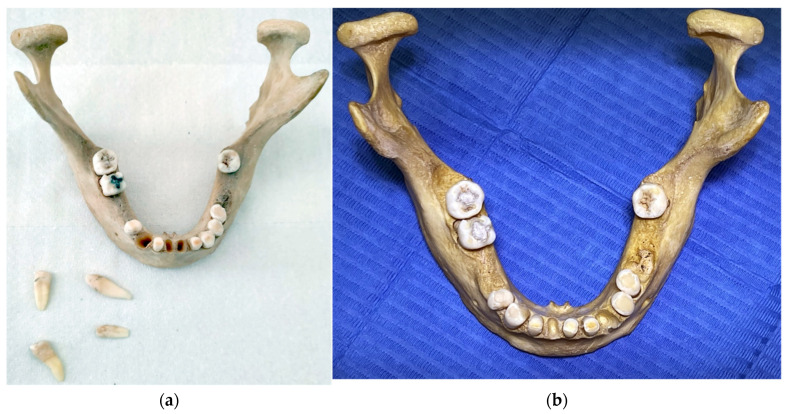
The human mandible with teeth: (**a**) before creating dental fillings; (**b**) after dental fillings and CBCT scan.

**Figure 2 molecules-27-04035-f002:**
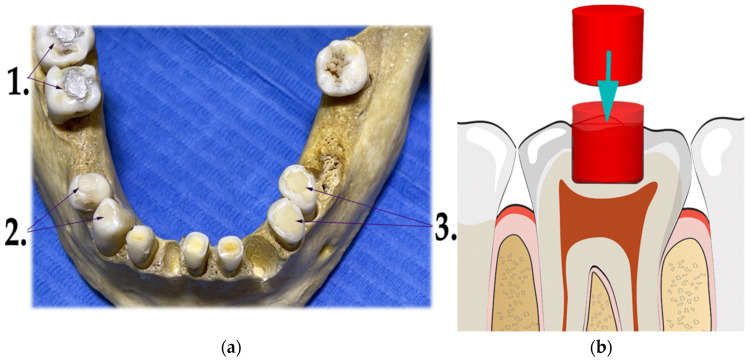
Dental fillings made on the teeth post-mortem. (**a**) Three dental materials were used in dental restorations: (1) amalgam extra-high copper non-gamma-two—Ana 2000 capsules, teeth 46 and 47; (2) glass ionomer—GC FUJI IX GP wear-resistant, teeth 43 and 44; (3) composite methacrylate resin—Neo Spectra ST (methacrylate-modified polysiloxane), teeth 34 and 35. (**b**) The filling was created with an occlusal part above the enamel surface with a horizontal plane with the intention to support the contact with the FTIR sensor.

**Figure 3 molecules-27-04035-f003:**
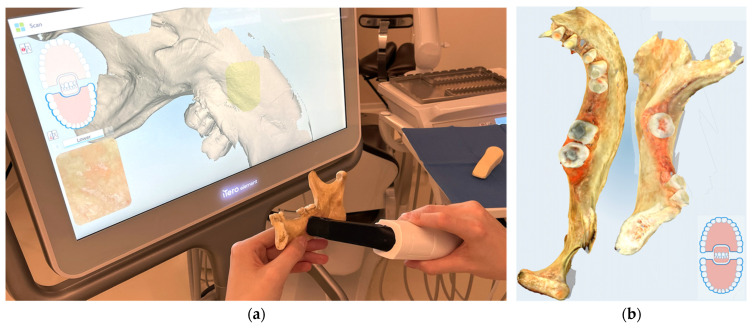
Optical scanning of the mandible: (**a**) iTero Element intraoral scanner (Align Technology, Inc., San Jose, California, USA) and other scanners are commonly available; (**b**) PLY/STL export for comparison to the CBCT-segmented models. This approach confirmed sufficient accuracy of the surface scan, albeit remains were optional, as the surface scan had no morphological information about the inner structure.

**Figure 4 molecules-27-04035-f004:**
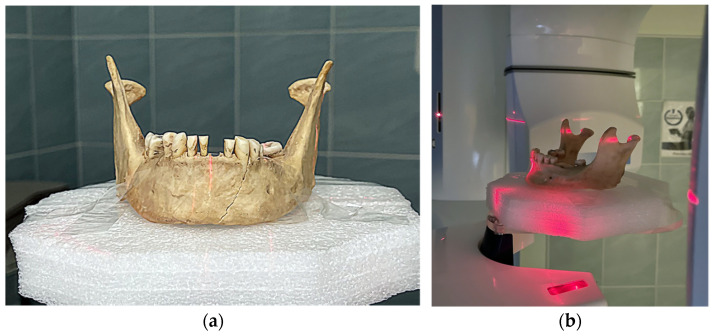
CBCT scanning with Planmeca ProMax 3D Mid CBCT (Planmeca Group, Helsinki, Finland): (**a**) Positioning of the mandible consisting of two fragments and stabilization of teeth in the sockets; (**b**) OPG and CBCT scans were created before and after dental restorations were made.

**Figure 5 molecules-27-04035-f005:**
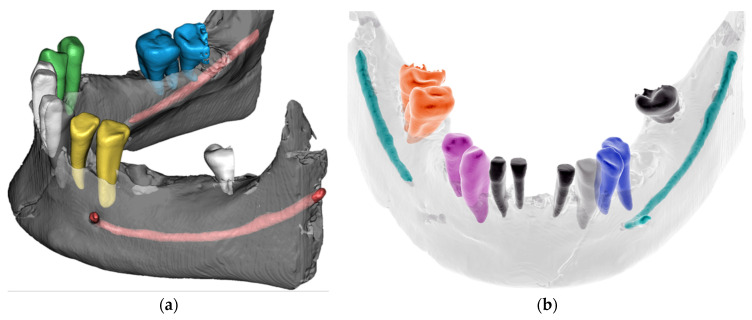
AI-automated segmentation of the mandible with the online tool Diagnocat: (**a**) Separate segments of teeth, bone, and nerve canal can be visualized with distinct colors, opacity and exported as separate STLs. (**b**) Scan of teeth without dental fillings shows unfinished recognition of the right mandibular canal and cut of distal parts of wisdom teeth.

**Figure 6 molecules-27-04035-f006:**
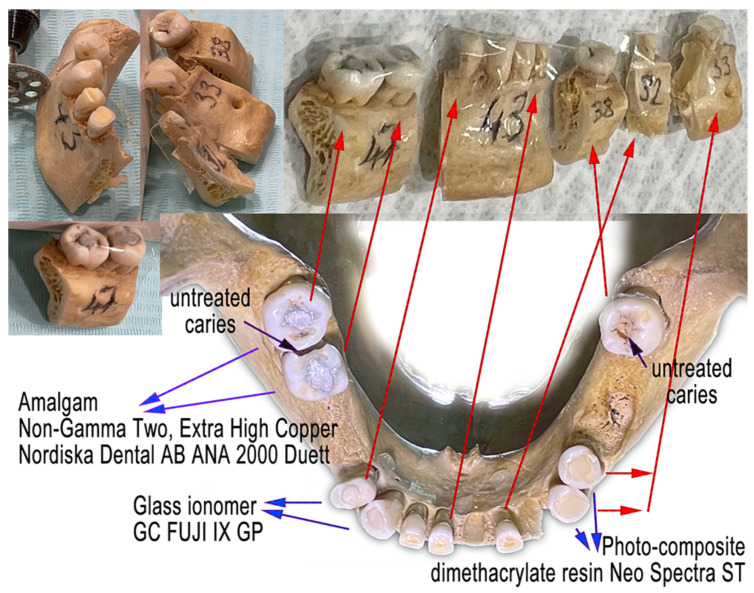
The mandible was cut into five samples and labeled (47, 43, 38, 32 and 33) to allow repeated sequences of sulfuric acid bathing, triple washing, drying, photographing and micro-CT scanning.

**Figure 7 molecules-27-04035-f007:**
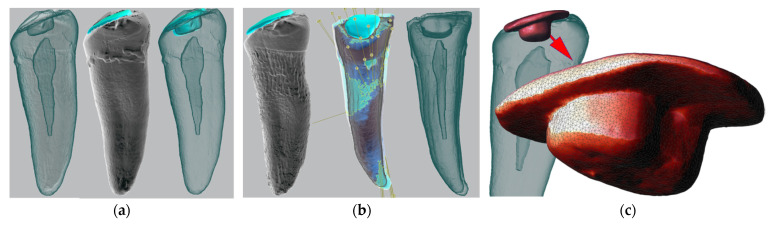
Outputs from micro-CT segmentation: (**a**) Original segment of tooth 34 before acid bath; (**b**) segments of tooth and dental filling after 96 h in acid bath; (**c**) 3D-reconstructed dental filling for future projections over various 2D (typically) diagnostic pre-mortem dental records (X-rays, digital intraoral photography).

**Figure 8 molecules-27-04035-f008:**
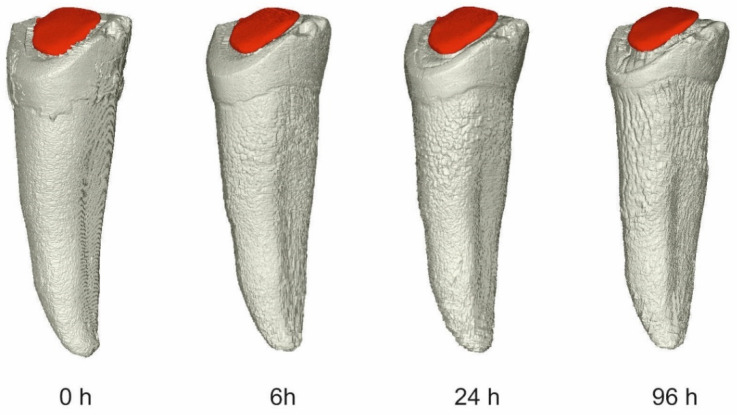
There were four micro-CT scans made for each sample, and each scan was segmented into three parts: bone, tooth, and in the case of the treated tooth, also with the dental material part. This picture shows a dental filling with composite resin on tooth 34 from the segment labeled “33” after exposure to 75% sulfuric acid after various times of exposure. It is obvious the composite filling is not losing volume in the acid bath, in contrast to the remaining tooth structures.

**Figure 9 molecules-27-04035-f009:**
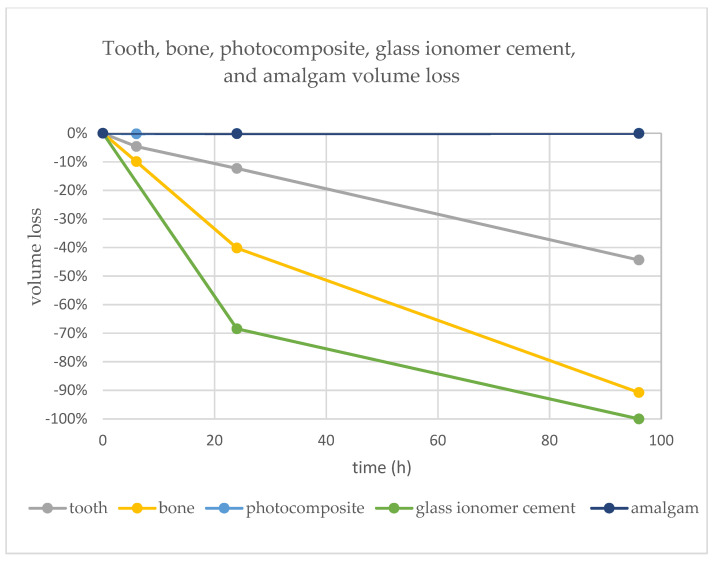
Graph representing volume loss over time in tooth, bone, composite, glass ionomer cement, and amalgam. The representation of composite (photocomposite) resin is hidden behind amalgam.

**Figure 10 molecules-27-04035-f010:**
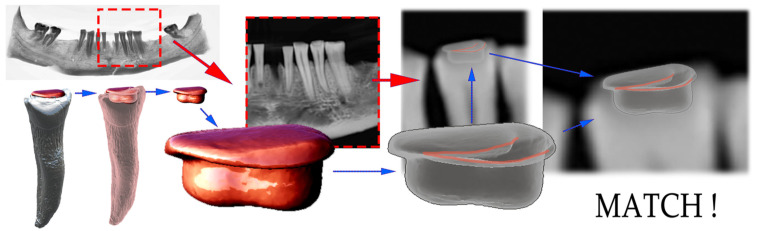
The reconstructed dental pattern of restoration of tooth 43 matched to 2D, representing volume loss over time in tooth, bone, composite, glass-ionomer cement, and amalgam. The representation of composite resin is hidden behind amalgam.

**Figure 11 molecules-27-04035-f011:**
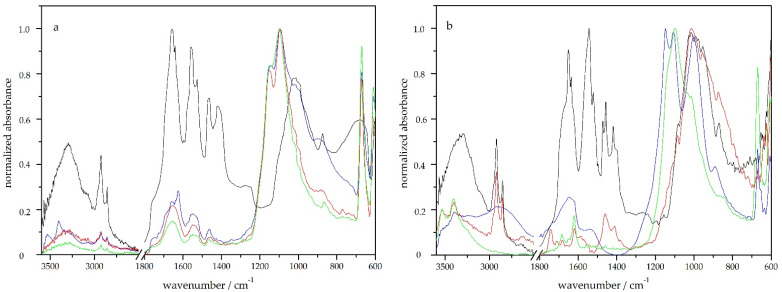
ATR-FTIR spectra of bone (**a**) and tooth (**b**) immersed in 75% sulfuric acid for 2 h (blue line), 6 h (red line), 24 h (green line) and original sample (black line).

**Table 1 molecules-27-04035-t001:** Volume loss of dental materials and bone after exposure to 75% sulfuric acid.

Time (h)	0	6	24	96
	Volume _0 h_ (mm^3^)	Volume _6 h_ (mm^3^)	Volume _24 h_ (mm^3^)	Volume _96 h_ (mm^3^)
sample 32				
dentine	190.29	185.1	170.5	110.95
percentage loss	0.0%	−2.7%	−10.4%	−41.7%
enamel	12.81	8.56	7.64	2.03
percentage loss	0.0%	−33.2%	−40.4%	−84.2%
tooth	203.1	193.7	178.12	112.98
percentage loss	0.0%	−4.6%	−12.3%	−44.4%
bone	668.1	601.85	399.56	61.77
percentage loss	0.0%	−9.9%	−40.2%	−90.8%
sample 33	324.45	299.16	290.64	270.57
	0.0%	−7.8%	−10.4%	−16.6%
Composite	16.46	16.43	16.45	16.46
	0.0%	−0.2%	−0.1%	0.0%
sample 43				
glass ionomer	11.57	^1^	3.65	0
	0.0%	^1^	−68.5%	−100.0%
sample 47				
amalgam	33.42	33.43	33.37	33.4
	0.0%	0.0%	−0.1%	−0.1%

^1^ No values for glass ionomer are a result of a drop-out of the tooth filling during the washing or drying process and skipping a round of micro-CT scanning. However, this was later found in the plastic sieves used in the process of washing.

## Data Availability

Data supporting the reported results are freely available at: https://drive.google.com/drive/folders/1UqhTbdDWZwxHamUMh6xxgqx3UDC1wimF, accessed on 15 June 2022. The online location provides a full digital photo set and STL segmentations of all four stages of degradation for bone and separately for teeth in that particular segment. Moreover, the digital intraoral scan of the mandible and the CT segmented with AI is provided at this location.

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
