# Peer review of "Human Remains Identification Using Micro-CT, Chemometric and AI Methods in Forensic Experimental Reconstruction of Dental Patterns after Concentrated Sulphuric Acid Significant Impact"

_molecules, 2022, doi:10.3390/molecules27134035_

Round 1
Reviewer 1 Report
The article investigates possibility of experimental reconstruction of the dental pattern after immersion of the specimen in concentrated sulphuric acid, by using CBCT, micro CT, chemometrics and artificial intelligence algorithms.
The experiment is interesting however the article is too long (33 pages in total) and not systematic, with much redundant information. Article cites 120 references, which is also too many to follow, use and read. The authors cite the same manuscript which is the object of the current review as the reference #15. Although it is published in Preprints, there is no justification for self-citation of the same article that is under review.
In the present state, the article is not readable nor clear. The article should be rewritten following the principles for writing scientific articles, that provide structure and clarity. Parts of the experiment presented in Materials & Methods should be presented in the same order and using the same terminology in Results, Discussion and Conclusion sections.
Some specific comments are given below.
Title: consider more precise “concentrated sulphuric acid” instead of “concentrated acid”.
Introduction
Too long: 10 pages! Too much of general information given, not focusing or introducing the readers to the present research, with repetition.
Dental pattern should be defined at the beginning of the Introduction section (presently lines 183-186).
Related to dental morphology analysis for identification purposes, inclusion of the following reference is suggested: G. Richard Scott ● Marin A. Pilloud ● David Navega ● João d’Oliveira Coelho ● Eugénia Cunha ● Joel D. Irish. rASUDAS A New Web-Based Application for Estimating Ancestry from Tooth Morphology. Forensic Anthropology Vol. 1, No. 1: 18–31. DOI 10.5744/fa.2018.0003.
Materials and Methods
Comments do not belong to the Materials and Methods section (lines 516, 631-636).
Only references that describe the methods (or sample) are to be cited (line 516).
Teeth should be described with FDI tooth notation system (line 547, description of the Figure 3 in lines 566-571).
Same terminology should be used throughout the article: line 598 “…mandible has been cut into five segments”, line 601 “five segments labelled 47, 42, 38, 32 and 33 (why 33 when this segment consists of teeth 34 and 35 with appropriate bone?). Table 1. describes “samples”. It should be either “segment” or “sample” throughout the article.
Authors first use the term “composite” (Abstract, Introduction, Material and Methods) than switch to “photo-composite” (Material and Methods, Results, Discussion, Conclusion).
Discussion
At the beginning of the Discussion section, the aim and findings of the current research should be summarized, commented and explained. Only after that, comparison with already published data should be made and discussion should lead to conclusion.
In the present manuscript, this rule was not followed, and authors discuss other research in extenso already from the 3rd sentence (from line 749 on). Own results are not clearly discussed.
Comparison between high silver and high copper amalgam was not the aim of the research, while authors cover this in text and also in Table 2.
Lines 763-766 should be edited for content and grammar.
Figures
Figure 1. a) should be enlarged so that details are visible to readers. It is not clear why some teeth are out alveoli? b) should be cut out as dentition is presented (after fillings) in Figure 3.
Author Response
Reviewer 1 - molecules-1729433
|
|
Can be improved |
Must be improved |
Not applicable |
|
|
Does the introduction provide sufficient background and include all relevant references? |
( ) |
( ) |
(x) |
( ) |
|
Are all the cited references relevant to the research? |
( ) |
( ) |
(x) |
( ) |
|
Is the research design appropriate? |
(x) |
( ) |
( ) |
( ) |
|
Are the methods adequately described? |
( ) |
(x) |
( ) |
( ) |
|
Are the results clearly presented? |
( ) |
(x) |
( ) |
( ) |
|
Are the conclusions supported by the results? |
(x) |
( ) |
( ) |
( ) |
Comments and Suggestions for Authors
Reviewer 1
Reviewer 1
Dear reviewer,
Thank you for your time and opinion. We incorporated your comments as best we could. Changes in the text and our answers are written below:
Moderate English changes required
The English was revised.
Comments and Suggestions for Authors:
The article investigates possibility of experimental reconstruction of the dental pattern after immersion of the specimen in concentrated sulphuric acid, by using CBCT, micro CT, chemometrics and artificial intelligence algorithms.
The experiment is interesting however the article is too long (33 pages in total) and not systematic, with much redundant information. Article cites 120 references, which is also too many to follow, use and read. The authors cite the same manuscript which is the object of the current review as the reference #15. Although it is published in Preprints, there is no justification for self-citation of the same article that is under review.
- Thank you for pointing out this error, the reference has been removed. We have now shortened the manuscript, summarized and citations were revised in all text. Redundant information has been mostly removed or some parts were moved to appendix that contains supplementary material that is not an essential part of the manuscript itself, albeit may be helpful in providing a more comprehensive understanding of the problem.
In the present state, the article is not readable nor clear. The article should be rewritten following the principles for writing scientific articles, that provide structure and clarity. Parts of the experiment presented in Materials & Methods should be presented in the same order and using the same terminology in Results, Discussion and Conclusion sections.
- All article parts were revised. Major parts of the manuscript were rewritten and are visualized with tracking changes – text in red. Some portions of the text were removed, others were reorganized to improve the structure, text flow and readability. Corresponding parts of the experiment were revised to be clearer, and terminology used to be coherent in corresponding sections.
Some specific comments are given below.
Title: consider more precise “concentrated sulphuric acid” instead of “concentrated acid”.
We added “sulphuric” to the title.
Introduction
Too long: 10 pages! Too much of general information given, not focusing or introducing the readers to the present research, with repetition.
- The introduction was shortened and is now better focused.
Dental pattern should be defined at the beginning of the Introduction section (presently lines 183-186).
- Dental pattern – the definition was added to introduction, on lines 95-103.
Related to dental morphology analysis for identification purposes, inclusion of the following reference is suggested: G. Richard Scott ● Marin A. Pilloud ● David Navega ● João d’Oliveira Coelho ● Eugénia Cunha ● Joel D. Irish. rASUDAS A New Web-Based Application for Estimating Ancestry from Tooth Morphology. Forensic Anthropology Vol. 1, No. 1: 18–31. DOI 10.5744/fa.2018.0003.
- Dental morphology analysis reference was added.
Materials and Methods
Comments do not belong to the Materials and Methods section (lines 516, 631-636).
- The comments were deleted.
Only references that describe the methods (or sample) are to be cited (line 516).
- References were edited in the whole article.
Teeth should be described with FDI tooth notation system (line 547, description of the Figure 3 in lines 566-571).
- This point was revised. FDI description has been added.
Same terminology should be used throughout the article: line 598 “…mandible has been cut into five segments”, line 601 “five segments labelled 47, 42, 38, 32 and 33 (why 33 when this segment consists of teeth 34 and 35 with appropriate bone?). Table 1. describes “samples”. It should be either “segment” or “sample” throughout the article.
- The terminology was revised.
Authors first use the term “composite” (Abstract, Introduction, Material and Methods) than switch to “photo-composite” (Material and Methods, Results, Discussion, Conclusion).
- The terminology “photo-composite” and “composite” were used synonymously which is clear to dentists with emphasis on either of process of making with light-curing or the general composition reference to keep the text simple. However, we have revised all the expressions used which are more coherent now.
Discussion
At the beginning of the Discussion section, the aim and findings of the current research should be summarized, commented and explained. Only after that, comparison with already published data should be made and discussion should lead to conclusion.
In the present manuscript, this rule was not followed, and authors discuss other research in extenso already from the 3rd sentence (from line 749 on). Own results are not clearly discussed.
- The sentences about results were added to the beginning of the Discussion. The whole chapter has been reorganized according to your recommendation.
Comparison between high silver and high copper amalgam was not the aim of the research, while authors cover this in text and also in Table 2.
- This result is also of interest for human forensic identification based on dental pattern reconstruction, as the materials are more durable than human tissue and the composition contributes to the length of acid degradation of the material.
Lines 763-766 should be edited for content and grammar.
- These lines were edited, and grammar was revised.
Figures
Figure 1. a) should be enlarged so that details are visible to readers. It is not clear why some teeth are out alveoli? b) should be cut out as dentition is presented (after fillings) in Figure 3.
- This picture has been provided to editor as high-definition picture. The current word document estimates overlap 200 MB which makes it difficult to upload and manipulate. The Figures in the current text - have been intentionally reduced in quality by higher compression. As the mandible has been destroyed during the experiment, it is not possible to make photo. This photo comes from forensic experts as the mandible was originally provided and some teeth are out of alveoli to visualize these were mobile and were later repositioned back into the alveolar bone.
Thank you for your time and useful remarks. We did our best to implement all your recomendations. All the changes are visualized in the text.

Reviewer 2 Report
In my opinion, it is an interesting paper.
I would like to suggest to the authors to reduce the extent of the introduction since it is huge. Moreover, the aim of the study should be presented in the abstract. The procedures are well presented through the figures. I think it is a very useful paper.
Author Response
Reviewer 2 molecules-1729433
|
|
Can be improved |
Must be improved |
Not applicable |
|
|
Does the introduction provide sufficient background and include all relevant references? |
( ) |
(x) |
( ) |
( ) |
|
Are all the cited references relevant to the research? |
(x) |
( ) |
( ) |
( ) |
|
Is the research design appropriate? |
(x) |
( ) |
( ) |
( ) |
|
Are the methods adequately described? |
(x) |
( ) |
( ) |
( ) |
|
Are the results clearly presented? |
(x) |
( ) |
( ) |
( ) |
|
Are the conclusions supported by the results? |
(x) |
( ) |
( ) |
( ) |
Comments and Suggestions for Authors
Comments and Suggestions for Authors:
In my opinion, it is an interesting paper.
I would like to suggest to the authors to reduce the extent of the introduction since it is huge. Moreover, the aim of the study should be presented in the abstract. The procedures are well presented through the figures. I think it is a very useful paper.
English language and style are fine/minor spell check required
Dear reviewer,
Thank you for your time and opinion. We incorporated your comments as best we could. We reduced introduction and we also revised English in our article.
All the changes are visualized in the text.
Submission Date
30 April 2022
Date of this review
22 May 2022 21:59:50
Round 2
Reviewer 1 Report
The manuscript is somewhat improved, however, it is still too long, especially Introduction section. It is understandable that of 10 authors each wanted to contribute, but there are still many redundant sections. The length of the text (30 pages, previously 33) and quantity of references (105, previously 120) are suitable for an extensive review paper or a book chapter. The present manuscript is a research article and needs to be concise and focused on the problems related to the aim of the research. Historical cases described could be used for another manuscript and submitted to some forensic odontology journal.
English proofread has been suggested, however, there are still sentences lacking some words (even in Highlights), punctuation errors (brackets in line 1112) etc. Term “dental preparation” has been used for “dental filling/restoration”, which is an error occurring throughout all sections of the manuscript.
Introduction section is suggested to be organized in six paragraphs (without titles) and not exceeding two pages: 1) teeth, dental pattern and forensic odontology, 2) influence of the acids on teeth and dental materials, 3) using CBCT and micro CT in dentistry and forensic science, 4) infrared spectroscopy in forensic dentistry, 5) artificial intelligence, and 6) aim of the present research. Parts of the content that authors regard important can be moved to the Discussion section.
Abstract
Line 31. As “tooth obliteration” is a term related to teeth with calcified pulp canals, another term is suggested: "dissolving". The suggestion is for the whole article.
Introduction
Lines 128-132 should be moved to the last paragraph of the Introduction section, without references, and followed by the aim. Consider replacing “we communicate” with “we have investigated”.
Lines 144-145 should be moved to Line 90 (“In the absence…).
Line 145 – there is contradictory information given: “Experience of other scientists (reference is missing here) shows that teeth are completely dissolved… while at 90 hours in 96% sulfuric acid.” and “Even after 90 hours in 96% sulfuric acid, the destruction of the samples was still incomplete”.
Line 163: The first sentence in the paragraph is stating the same as the second, so one should be deleted. The second sentence should be edited for grammar and content.
Line 626, Highlights section, some words are missing in the sentence.
Reference 49 (lines 222, 1202) is used in the context with CBCT, while it is a study conducted on OPGs.
Materials and Methods
It was stressed in the first-round review that only the references that describe the methods (or sample) are to be cited. There are still references in the sentence with hypothesis (line 636), which should be the idea of the authors that is to be tested in the experiment.
Lines 697, 718 – Consider using plural instead of singular in the sentence (“Old dental fillings were removed.”), line 718 “fillings” instead of “filling”.
Line 719 – The last sentence in the figure legend should be deleted, as information has been given already in the text (lines 699, 700).
It is suggested again that either “photocomposite” or “composite” should be used throughout the text, not both. Consistency is important for clarity and readability of the text.
Results
Table 2. is not presenting authors’ results so should be deleted.
Lines 992,993 – “This is possibly first implementation of advanced AI algorithm utilization in clinical forensic research”. The present research was not clinical, yet experiment conducted in the laboratory.
Figure 12.(a) is presenting equipment (FTIR spectrophotometer) and not result, so should be deleted.
Discussion
Lines 1126-1133 reference is missing, also the year of the conference.
Lines 1133-1135 general information is given, but not explained how it was done in the described experiment without consequences.
Author Response
Reviewer 1 - Review Report (Round 2)
Open Review
(x) I would not like to sign my review report
English language and style
(x) Moderate English changes required
|
Yes |
Can be improved |
Must be improved |
Not applicable |
|
|
Does the introduction provide sufficient background and include all relevant references? |
( ) |
( ) |
(x) |
( ) |
|
Are all the cited references relevant to the research? |
( ) |
( ) |
(x) |
( ) |
|
Is the research design appropriate? |
(x) |
( ) |
( ) |
( ) |
|
Are the methods adequately described? |
( ) |
(x) |
( ) |
( ) |
|
Are the results clearly presented? |
( ) |
(x) |
( ) |
( ) |
|
Are the conclusions supported by the results? |
(x) |
( ) |
( ) |
( ) |
Comments and Suggestions for Authors
The manuscript is somewhat improved, however, it is still too long, especially Introduction section. It is understandable that of 10 authors each wanted to contribute, but there are still many redundant sections. The length of the text (30 pages, previously 33) and quantity of references (105, previously 120) are suitable for an extensive review paper or a book chapter. The present manuscript is a research article and needs to be concise and focused on the problems related to the aim of the research. Historical cases described could be used for another manuscript and submitted to some forensic odontology journal.
- Dear reviewer, thank you for your time and effort to help us improve our manuscript.
- We have respected your advice and we have significantly reduced the manuscript length with focus on Introduction chapter, which is now distributed in 6 paragraphs, without subchapters with focus you have recommended.
- The introduction chapter has been significantly reduced to little above 2 pages. (From 10 pages in version after 1st round of revisions).
- Historical context has been removed so were the redundant sections. The length of the text without Appendix (15 pages, previously 33) and quantity of references (83, previously 120)
English proofread has been suggested, however, there are still sentences lacking some words (even in Highlights), punctuation errors (brackets in line 1112) etc. Term “dental preparation” has been used for “dental filling/restoration”, which is an error occurring throughout all sections of the manuscript.
- English was revised and also the term was changed.
Introduction section is suggested to be organized in six paragraphs (without titles) and not exceeding two pages: 1) teeth, dental pattern and forensic odontology, 2) influence of the acids on teeth and dental materials, 3) using CBCT and micro CT in dentistry and forensic science, 4) infrared spectroscopy in forensic dentistry, 5) artificial intelligence, and 6) aim of the present research. Parts of the content that authors regard important can be moved to the Discussion section.
- Chapter titles were removed. Few relevant parts from Introduction were moved to Discussion chapter, majority of indirect context was removed. The manuscript is now more focused and concise.
Abstract
Line 31. As “tooth obliteration” is a term related to teeth with calcified pulp canals, another term is suggested: "dissolving". The suggestion is for the whole article.
- “tooth oblieration” was corrected to "dissolving”.
Introduction
Lines 128-132 should be moved to the last paragraph of the Introduction section, without references, and followed by the aim. Consider replacing “we communicate” with “we have investigated”.
- This point was revised.
Lines 144-145 should be moved to Line 90 (“In the absence…).
- We moved the line.
Line 145 – there is contradictory information given: “Experience of other scientists (reference is missing here) shows that teeth are completely dissolved… while at 90 hours in 96% sulfuric acid.” and “Even after 90 hours in 96% sulfuric acid, the destruction of the samples was still incomplete”.
- Reference was added, the point was revised.
Line 163: The first sentence in the paragraph is stating the same as the second, so one should be deleted. The second sentence should be edited for grammar and content.
- the point was revised.
Line 626, Highlights section, some words are missing in the sentence.
- the point was revised.
Reference 49 (lines 222, 1202) is used in the context with CBCT, while it is a study conducted on OPGs.
- the point was revised.
Materials and Methods
It was stressed in the first-round review that only the references that describe the methods (or sample) are to be cited. There are still references in the sentence with hypothesis (line 636), which should be the idea of the authors that is to be tested in the experiment.
- the point was revised.
Lines 697, 718 – Consider using plural instead of singular in the sentence (“Old dental fillings were removed.”), line 718 “fillings” instead of “filling”.
- This was revised.
Line 719 – The last sentence in the figure legend should be deleted, as information has been given already in the text (lines 699, 700).
- The sentence was deleted.
It is suggested again that either “photocomposite” or “composite” should be used throughout the text, not both. Consistency is important for clarity and readability of the text.
- This was revised, term “composite” was preferred.
Results
Table 2. is not presenting authors’ results so should be deleted.
- Table 2 was deleted.
Lines 992,993 – “This is possibly first implementation of advanced AI algorithm utilization in clinical forensic research”. The present research was not clinical, yet experiment conducted in the laboratory.
- The sentence was corrected.
Figure 12.(a) is presenting equipment (FTIR spectrophotometer) and not result, so should be deleted.
- The figure was deleted.
Discussion
Lines 1126-1133 reference is missing, also the year of the conference.
- This point was revised, and the reference has been added.
Lines 1133-1135 general information is given, but not explained how it was done in the described experiment without consequences.
- This point was revised, and the reference has been added.
Dear reviewer,
Thank you for your feedback. We have revised the manuscript as best we can, and we hope that it will meet your expectations and requirements.
